# Mortality and major disease risk among migrants of the 1991–2001 Balkan wars to Sweden: A register-based cohort study

**Edda Bjork Thordardottir**[1], **Li Yin**[2], **Arna Hauksdottir**[1], **Ellenor Mittendorfer-Rutz**[3], **Anna-Clara Hollander**[4], **Christina M. Hultman**[2,5], **Paul Lichtenstein**[2], **Weimin Ye**[2], **Filip K. Arnberg**[6,7], **Fang Fang**[8], **Emily A. Holmes**[3,9‡], **Unnur Anna Valdimarsdottir**[1,2,10‡]*

1 Center of Public Health Sciences, University of Iceland, Reykjavik, Iceland, 2 Department of Medical Epidemiology and Biostatistics, Karolinska Institutet, Stockholm, Sweden, 3 Division of Insurance Medicine, Department of Clinical Neuroscience, Karolinska Institutet, Stockholm, Sweden, 4 Department of Public Health Sciences, Karolinska Institutet, Stockholm, Sweden, 5 Icahn School of Medicine, Mount Sinai Hospital, New York, New York, United States of America, 6 Department of Neuroscience, Uppsala Universitet, Uppsala, Sweden, 7 Stress Research Institute, Stockholm University, Stockholm, Sweden, 8 Institute of Environmental Medicine, Karolinska Institutet, Stockholm, Sweden, 9 Department of Psychology, Uppsala Universitet, Uppsala, Sweden, 10 Department of Epidemiology, Harvard T.H. Chan School of Public Health, Boston, Massachusetts, United States of America

‡ These authors are joint senior authors on this work.
* unnurav@hi.is

**Data Availability Statement:** Data is available on request for any interested researchers to allow

## Abstract

### Background

In recent decades, millions of refugees and migrants have fled wars and sought asylum in Europe. The aim of this study was to quantify the risk of mortality and major diseases among migrants during the 1991–2001 Balkan wars to Sweden in comparison to other European migrants to Sweden during the same period.

### Methods and findings

We conducted a register-based cohort study of 104,770 migrants to Sweden from the former Yugoslavia during the Balkan wars and 147,430 migrants to Sweden from 24 other European countries during the same period (1991–2001). Inpatient and specialized outpatient diagnoses of cardiovascular disease (CVD), cancer, and psychiatric disorders were obtained from the Swedish National Patient Register and the Swedish Cancer Register, and mortality data from the Swedish Cause of Death Register. Adjusting for individual-level data on sociodemographic characteristics and emigration country smoking prevalence, we used Cox regressions to contrast risks of health outcomes for migrants of the Balkan wars and other European migrants. During an average of 12.26 years of follow-up, being a migrant of the Balkan wars was associated with an elevated risk of being diagnosed with CVD (HR 1.39, 95% CI 1.34–1.43, $p < 0.001$) and dying from CVD (HR 1.45, 95% CI 1.29–1.62, $p < 0.001$), as well as being diagnosed with cancer (HR 1.16, 95% CI 1.08–1.24, $p < 0.001$) and dying from cancer (HR 1.27, 95% CI 1.15–1.41, $p < 0.001$), compared to other European migrants. Being a migrant of the Balkan wars was also associated with a greater overall risk of being diagnosed with a psychiatric disorder (HR 1.19, 95% CI 1.14–1.23, $p < 0.001$),

replication of results through the Swedish National Data Service, provided all ethical and legal requirements are met. Detailed information on data application can be found at https://www.registerforskning.se/en/. Registries used for this study include the Swedish Cancer Registry, the Swedish Causes of Death Register, the Total Population Register, the Swedish National Patient Register, the Swedish Education Registry and the Multi-Generation Register.

**Funding:** EBT is supported by the Icelandic Research Fund (grant no. 185287-051) (website: https://en.rannis.is/). UAV reports grants from the Grant of Excellence, Icelandic Research Fund (grant no. 163362-051), and ERC Consolidator Grant (StressGene, grant no: 726413). EH reports grants from The Lupina Foundation, Swedish Research Council (2017-00957), and The Oak Foundation (OCAY-18-442). The funders had no role in study design, data collection and analysis, decision to publish, or preparation of the manuscript.

**Competing interests:** The authors have declared that no competing interests exist.

**Abbreviations:** CVD, cardiovascular disease; PTSD, post-traumatic stress disorder.

particularly post-traumatic stress disorder (HR 9.33, 95% CI 7.96–10.94, $p < 0.001$), while being associated with a reduced risk of suicide (HR 0.68, 95% CI 0.48–0.96, $p = 0.030$) and suicide attempt (HR 0.57, 95% CI 0.51–0.65, $p < 0.001$). Later time period of migration and not having any first-degree relatives in Sweden at the time of immigration were associated with greater increases in risk of CVD and psychiatric disorders. Limitations of the study included lack of individual-level information on health status and behaviors of migrants at the time of immigration.

## Conclusions

Our findings indicate that migrants of the Balkan wars faced considerably elevated risks of major diseases and mortality in their first decade in Sweden compared to other European migrants. War migrants without family members in Sweden or with more recent immigration may be particularly vulnerable to adverse health outcomes. Results underscore that persons displaced by war are a vulnerable group in need of long-term health surveillance for psychiatric disorders and somatic disease.

## Author summary

### Why was this study done?

- Understanding the toll of war on the health of migrants is a highly relevant and pressing issue in light of the global humanitarian crisis, with more people than ever affected by forced displacement.

- The 1991–2001 Balkan wars were marked by war crimes such as genocide, ethnic cleansing, rape, and torture.

- More than 100,000 adults and children migrated to Sweden from the former Yugoslavia during the Balkan wars.

### What did the researchers do and find?

- Using Swedish registry data, we assessed morbidity and mortality among 104,770 migrants to Sweden from former Yugoslavia during the Balkan wars and 147,430 migrants from 24 other European countries immigrating to Sweden from 1991 to 2001.

- We found that compared to other European migrants, being a migrant of the Balkan wars was associated with an elevated risk of overall psychiatric disorders, particularly post-traumatic stress disorder, along with a reduced risk of suicide and suicide attempt.

- Being a migrant of the Balkan wars was also associated with being diagnosed with and dying from both cancer and cardiovascular disease.

- Later migration to Sweden and having no first-degree relatives in Sweden at immigration was associated with the greatest risk elevation of psychiatric disorders and cardiovascular disease among migrants fleeing armed conflict.

**What do these findings mean?**

- These findings indicate that being a war migrant may be associated with considerable elevations in post-traumatic stress disorder as well as risks of cardiovascular- and cancer-related morbidities and mortality, particularly among those migrating late in the wars and without family members.

- War migrants are a particularly vulnerable group that health professionals should monitor over the long term.

- Host countries should make availability and accessibility of healthcare and social services to war migrants a priority.

- Potential limitations of this study include lack of information about health status at the time of immigration as well as behavioral factors possibly contributing to increased disease risk such as smoking and alcohol consumption.

## Introduction

The global population of people forced to cross national boundaries due to war, persecution, or violence reached 25.9 million at the end of 2018, representing the highest levels of external displacement ever recorded [1]. Due to this unprecedented influx of refugees and migrants to new countries, particularly from the Syrian Arab Republic, it is imperative to understand the extent of health consequences experienced by war migrant populations over the years after they arrive in a host country.

All migrants face resettlement stressors, regardless of the reason for migration, including socioeconomic adversities and accommodating to a new language and culture. Beyond these resettlement stressors, migrants of war can carry the additional burden of exposures to traumatic events such as witnessing or experiencing the threat of death, torture, violence, and persecution before or during their journey to a host country. Earlier studies have reported increased risk of psychiatric morbidity, such as post-traumatic stress disorder (PTSD), depression, and psychosis, among war migrants when compared to population natives [2,3]. However, as general stressors associated with migration are not accounted for in this comparison, we can better understand how war migration affects psychiatric morbidity if we use other migrants as a reference group. Meanwhile, few studies have assessed the risk of physical morbidity and mortality among war migrants, and results have been conflicting. Some studies indicate a decreased risk of cancer and cardiovascular disease (CVD) among migrants compared to population natives [4] while others have found migrants to be at increased risk [5,6]. To date, the majority of studies on the health of war migrants rely on comparison to population natives. This comparison is subject to the so-called healthy migrant effect, because relative to the native population, the prevalence of most frequent diseases is lower among migrants [7].

The 1991–2001 Balkan wars were marked by war crimes such as genocide, ethnic cleansing, rape, and torture. The wars are estimated to have taken 140,000 lives, and led to the displacement of approximately 4 million people [8]. During the wars, more than 100,000 adults and children migrated from the Balkans to Sweden. Extending previous research on the health impact of war migration, we here utilize the Swedish national health registries to quantify mortality and major disease risk among migrants of the Balkan wars to Sweden, as well as risk factors for adverse outcomes.

## Methods

### Study population and design

We conducted a historical register-based cohort study in a number of Swedish national registries, using the personal identity numbers that are uniquely assigned to every resident at birth or immigration [9]. Through the Total Population Register we identified all migrants (*n* = 104,770) entering Sweden from 1 January 1991 through 31 December 2001 with a country of birth registered as Albania, Bosnia-Herzegovina, Croatia, Macedonia, Slovenia, or Yugoslavia. As we had no information on individual legal status at immigration, we adhere to the United Nations Migration Agency's definition of a migrant [10] and hereafter refer to these individuals as migrants of the Balkan wars. For comparison, we identified a cohort of all migrants from other European countries (*n* = 147,430) entering Sweden during the same time period with a country of birth registered as Austria, Belgium, Czech Republic, Czechoslovakia, Denmark, Finland, France, Germany, Great Britain, Greece, Hungary, Iceland, Ireland, Italy, Malta, Moldavia, the Netherlands, Norway, Poland, Portugal, Romania, Slovak Republic, Spain, or Switzerland (see S1 Table for number of migrants from each country). These countries were not at war during the specified time period.

Of the 252,200 identified migrants, 34 were excluded from analysis due to inconsistency of information concerning, for example, the date of death. The total number of migrants in the study was therefore 252,166. We followed these migrants for major disease outcomes (specified below) from the date of immigration until death, emigration out of Sweden, or end of follow-up (31 December 2010), whichever occurred first.

### Ethics statement

The overarching study protocol (S1 Study Protocol) was submitted and approved by the Regional Ethics Committee in Stockholm (nr. 2016/384-31), and the study is reported as per the Strengthening the Reporting of Observational Studies in Epidemiology (STROBE) guideline (S1 Text STROBE Checklist).

### Outcomes

Mortality outcomes during follow-up were identified in the Swedish Cause of Death Register and based on the International Classification of Diseases versions 9 and 10 (ICD-9 and ICD-10); we recorded overall and cause-specific mortality, including death due to CVD (ICD-9: 390–489; ICD-10: I00–I99), cancer (ICD-9: 140–209; ICD-10: C00–C99), suicide (ICD-9: 950–959; ICD-10: X60–X84), and other causes. The Swedish Cause of Death Register has been found to be a largely complete and high-quality data source [11].

The Swedish Cancer Register was utilized to obtain all newly diagnosed cancers (ICD-7: 140–209), and the Swedish National Patient Register was utilized to obtain inpatient and specialized outpatient diagnoses of CVD (ICD-9: 390–489; ICD-10: I00–199), suicide attempt (ICD 9: 950–958, 980–988; ICD 10: X60–X84, Y10–Y34), and psychiatric disorders (ICD-9: 290–319; ICD-10: F00–F99). The Swedish National Patient Register has been reported to have high validity, both for CVD and psychiatric disorders [12]. As it is likely that the migrants of the Balkan wars were exposed to multiple traumatizing events [8], we specifically looked at PTSD (ICD-9: 309B; ICD-10: F43.1). The validity of PTSD diagnoses in Swedish register data has been found to be sufficient, with a positive predictive validity of 76%–90% [13].

We conducted, ad hoc, sub-analyses on (1) smoking-related cancers, defined as buccal (ICD-7: 140–148), esophagus (ICD-7: 150), stomach (ICD-7: 151), large intestine (ICD-7: 153), pancreatic (ICD-7: 157), lung (ICD-7: 162–163), uterus (ICD-7: 171), and kidney cancer

including renal pelvis (ICD-7: 180); (2) alcohol-related cancers, defined as buccal (ICD-7: 140–148), digestive (ICD-7: 150), liver (primary site) (ICD-7: 155), large intestine and rectum (ICD-7: 153–154), larynx (ICD-7: 161), and breast cancer (ICD-7: 170); and (3) other cancers, i.e., any cancer excluding the above.

## Covariates

Covariates included age at immigration, sex, calendar period of immigration (1991–1994, 1995–1998, or 1999–2001), and educational attainment. The Multi-Generation Register was used to identify the number of first-degree relatives (i.e., siblings, parents, and children) also registered in Sweden at the time of immigration to the country (i.e., both migrating with the individual and already residing in Sweden).

Smoking is a major risk factor for morbidities and premature mortality, and may be unequally distributed between the migrants of the Balkan wars and other European immigrant groups. We therefore, ad hoc, obtained an estimated smoking prevalence for each migrant's home country from the World Health Organization's Global Health Observatory data repository for the year 2000 (S2 Table). As information was unavailable for Macedonia in the data repository, data from 2009 was used. Smoking prevalence was obtained for both sexes, for everyone 15 years and older, and was defined as daily or occasional use of tobacco, including cigarettes, cigars, pipes, or any other smoked tobacco products. When adjusting for smoking in the multivariable models, each individual received the smoking status of the country they emigrated from.

In addition, as we were concerned for differential baseline risks across populations, we subsequently adjusted for country-level cancer- and CVD-related death rates. We obtained death rates for migrants' home countries from the WHO Global Health Observatory data repository for the year 1990 (S3 Table).

In order to understand if suicide rates differed between the populations in the countries of the Balkan wars and other European countries, we obtained data on mean suicide rates for all studied countries for the year 1990 from Our World in Data [14] (S4 Table).

## Statistical analysis

Descriptive analyses were conducted to examine baseline factors across the 2 migrant populations, including sex, age, period of immigration, educational attainment (missing information was presented in a separate category), number of first-degree relatives in Sweden, and estimated smoking prevalence for each migrant's home country. *T* tests and chi-squared tests were conducted to determine if the 2 groups differed with respect to these characteristics.

Incidence rates (per 100,000 person-years) of morbidities and mortality during the time intervals 0–1, 2–4, 5–9, and 10 and more years since migration were calculated among migrants of the Balkan wars and other European migrants. Cox proportional hazard regression models were used to assess the hazard ratios (HRs) and 95% confidence intervals (CIs) of mortality and morbidities, comparing disease incidence rates and mortality rates in the 2 migrant populations. The proportionality assumption for all outcomes was not violated in any of the Cox models except for the analysis of PTSD; a model satisfying the assumption was complex but did not change the pattern of an overall high HR for PTSD. We therefore report all estimates by time intervals (0–1, 2–4, 5–9, and ≥10 years).

We present incidence rates and HRs first stratified by background characteristics (e.g., age at immigration, sex, education, period of immigration, and the number of first-degree relatives in Sweden at immigration), then age-adjusted HRs; next HRs adjusted for age at immigration,

sex, education, and calendar period of immigration; and finally HRs adjusted additionally for country-specific smoking prevalence.

In tables presenting rates of morbidities and mortality by time since migration, we adjusted all models for sex, education, age at immigration, smoking (country-specific prevalence), and calendar period of immigration. Age at immigration, calendar period of immigration, and smoking were modeled as continuous variables in all tables. When assessing death due to CVD, we adjusted for the CVD-related death rate of the home country. Similarly, when assessing death due to cancer we adjusted for the cancer-related death rate of the home country. Finally, linear regression analysis was conducted to test if HRs for total mortality, CVD mortality and morbidity, psychiatric morbidity, and PTSD were higher among those with (1) late period of immigration (1999–2001) and (2) no first-degree relatives in Sweden. For the statistical analyses, we used SAS software, version 9.2 (SAS Institute).

## Results

Migrants of the Balkan wars were more likely than other European migrants to be younger, to be female, and to have migrated to Sweden in an early time period (1991–1994) ($p < 0.001$). Mean age at immigration was slightly lower among migrants of the Balkan wars than among other European migrants, 28.02 versus 28.64 years, respectively. Median follow-up time was significantly longer for migrants of the Balkan wars (5,987 days) than for other European migrants (3,405 days). Migrants of the Balkan wars were less likely to have a university-level education and more likely to have 1 or more first-degree relatives in Sweden at the time of immigration to Sweden, compared to other European migrants ($p < 0.001$). The average country prevalence of smoking in the countries of the Balkan wars was higher than in the other European countries (46.55% versus 35.70%, $p < 0.001$) (Table 1).

The total follow-up for mortality was 1,531,938 person-years for migrants of the Balkan wars and 1,308,877 person-years for migrants of other European countries.

### Mortality

Table 2 shows mortality rates among migrants of the Balkan wars and other European migrants stratified by background characteristics as well as crude, age-adjusted, and multivariable-adjusted HRs for overall mortality, suicide, and mortality due to CVD and cancer. The crude hazard ratios varied considerably across strata of background factors without a clear pattern except across the number of first-degree relatives.

Among migrants of the Balkan wars, having a lower number of first-degree relatives in Sweden at immigration was associated with a more pronounced risk increase for overall mortality (HR 2.00, 95% CI 1.90–2.11, $p < 0.001$, for no relatives; HR 0.67, 95% CI 0.62–0.72, $p < 0.001$, for 1 relative; HR 0.49, 95% CI 0.39–0.61, $p < 0.001$, for 2 or more relatives) and death due to CVD (HR 2.33, 95% CI 2.13–2.55, $p < 0.001$, for no relatives; HR 0.56, 95% CI 0.49–0.64, $p < 0.001$, for 1 relative; HR 0.16, 95% CI 0.08–0.34, $p < 0.001$, for 2 or more relatives) (Table 2). With a trend test, total mortality ($p < 0.001$) and CVD mortality ($p < 0.001$) decrease significantly with the number of first-degree relatives.

Compared to other European migrants, being a migrant of the Balkan wars was associated with an elevated risk of overall mortality, with risk estimates increasing in multivariable-adjusted models (HR 1.20, 95% CI 1.14–1.27, $p < 0.001$) compared to crude data (HR 1.12, 95% CI 1.07–1.16, $p < 0.001$). Similarly, being a migrant of the Balkan wars was associated with an elevated risk of CVD mortality, with risk estimates increasing in multivariable-adjusted models (HR 1.45, 95% CI 1.29–1.62, $p < 0.001$) compared to the crude model (HR 1.20, 95% CI 1.11–1.29, $p < 0.001$). Elevated risk of cancer mortality was also associated with

**Table 1. Descriptive characteristics of migrants of the Balkan wars and other European migrants to Sweden during 1991–2001.**

| Characteristic | War migrants | Other European migrants | p-Value* |
|---|---|---|---|
| **Total** | 104,770 | 147,430 | |
| **Sex female** | 52,337 (49.95) | 70,249 (47.66) | <0.001 |
| **Age** | | | <0.001 |
| 0–19 years | 36,146 (34.50) | 33,578 (22.78) | |
| 20–39 years | 44,805 (42.77) | 82,685 (56.10) | |
| 40–59 years | 16,619 (15.86) | 23,811 (16.15) | |
| 60–79 years | 6,834 (6.52) | 6,706 (4.55) | |
| 80 years or older | 365 (0.35) | 617 (0.42) | |
| **Mean age at immigration (years)** | 28.02 | 28.64 | |
| **Mean number of follow-up (days)** | 5,987 | 3,405 | <0.001 |
| **Period of migration** | | | <0.001 |
| 1991–1994 | 70,857 (67.63) | 41,592 (28.22) | |
| 1995–1998 | 19,625 (18.73) | 42,636 (28.93) | |
| 1999–2001 | 14,287 (13.64) | 63,169 (42.86) | |
| **Education** | | | <0.001 |
| Primary | 39,756 (39.33) | 63,104 (49.81) | |
| Secondary | 45,682 (45.20) | 28,340 (22.37) | |
| University | 15,639 (15.47) | 35,257 (27.82) | |
| *Missing* | *3,692* | *20,696* | |
| **Number of first-degree relatives in Sweden at immigration** | | | <0.001 |
| 0 | 40,689 (38.84) | 87,365 (59.27) | |
| 1 | 32,045 (30.59) | 34,252 (23.24) | |
| 2+ | 32,035 (30.58) | 25,780 (17.49) | |
| **Smoking (%)**** | 46.55 | 35.70 | <0.001 |

Data are given as *n* (percent) unless otherwise indicated.

*T test for continuous variables: age at immigration, follow-up time, smoking; chi-squared test for other (categorical) variables.

**Estimated smoking prevalence of the migrant's home country, according to the World Health Organization's Global Health Observatory data repository for the year 2000.

being a migrant of the Balkan wars, with risk estimates somewhat attenuated in the multivariable-adjusted models (HR 1.27, 95% CI 1.15–1.41, $p < 0.001$) compared to the crude model (HR 1.36, 95% CI 1.26–1.47, $p < 0.001$) (Table 2).

Interestingly, being a migrant of the Balkan wars was associated with a decreased mortality risk due to suicide during follow-up, with similar risk estimates in crude (HR 0.69, 95% CI 0.54–0.89, $p = 0.004$) and multivariable-adjusted models (HR 0.68, 95% CI 0.48–0.96, $p = 0.030$).

When assessing time since immigration, we found that the risk of overall mortality associated with being a migrant of the Balkan wars was elevated at 5–9 years (HR 1.20, 95% CI 1.09–1.32, $p < 0.001$) and ≥10 years after immigration (HR 1.35, 95% CI 1.22–1.48, $p < 0.001$). Being a migrant of the Balkan wars was also associated with a gradual increase in risk of death due to CVD over time (HR 1.38, 95% CI 1.05–1.82, $p = 0.021$, at 2 to 4 years after immigration versus HR 1.55, 95% CI 1.30–1.86, $p < 0.001$, at ≥10 years after immigration). The risk of death due to cancer associated with being a migrant of the Balkan wars was increased throughout follow-up. Although not statistically significant, the risk of suicide associated with being a migrant of the Balkan wars was decreased at all time points (Table 3).

**Table 2. Mortality rates in migrants of the Balkan wars compared to other European migrants, across strata of sociodemographic and lifestyle-related characteristics.**

| Cause of death | Analysis or characteristic | IR* (95% CI) for war migrants | IR (95% CI) for other European migrants | HR (95% CI) | p-Value for HR |
|---|---|---|---|---|---|
| Overall mortality | **Univariate analysis** | | | | |
| | Crude overall HR | 348.8 (339.5–358.3) | 291.0 (281.9–300.4) | 1.12 (1.07–1.16) | <0.001 |
| | Age at immigration | | | | |
| | 0–19 years | 25.24 (21.39–29.79) | 24.09 (19.21 30.21) | 1.02 (0.76–1.35) | 0.917 |
| | 20–39 years | 85.75 (78.98–93.11) | 64.78 (59.13–70.98) | 1.19 (1.05–1.35) | 0.007 |
| | 40–59 years | 660.15 (628.20–693.72) | 480.65 (452.58–510.46) | 1.22 (1.12–1.32) | <0.001 |
| | 60–79 years | 3,639.37 (3,507.27–3,776.46) | 2,835.20 (2,705.39–2,971.24) | 1.20 (1.13–1.27) | <0.001 |
| | 80 years or older | 10,783.76 (9,560.49–12,163.54) | 13,006.87 (11,874.45–14,247.29) | 0.81 (0.70–0.95) | 0.007 |
| | Sex | | | | |
| | Male | 350.23 (337.25–363.71) | 315.96 (302.68–329.82) | 1.03 (0.97–1.09) | 0.034 |
| | Female | 347.30 (334.33–360.78) | 265.69 (253.44–278.53) | 1.22 (1.15–1.30) | <0.001 |
| | Education | | | | |
| | Primary | 686.50 (664.98–708.71) | 527.75 (507.08–549.27) | 1.18 (1.12–1.24) | <0.001 |
| | Secondary | 139.54 (131.13–148.48) | 213.19 (198.27–229.24) | 0.61 (0.55–0.67) | <0.001 |
| | University | 167.19 (151.57–184.42) | 94. 33 (85.06–104.61) | 1.57 (1.36–1.82) | <0.001 |
| | Missing | 555.17 (475.71–647.91) | 241.88 (216.63–270.07) | 2.43 (2.01–2.94) | <0.001 |
| | Period of immigration | | | | |
| | 1991–1994 | 297.96 (288.10–308.15) | 356.23 (338.95–374.39) | 0.82 (0.77–0.87) | <0.001 |
| | 1995–1998 | 639.81 (609.58–671.53) | 324.51 (306.64–343.42) | 1.92 (1.78–2.07) | <0.001 |
| | 1999–2001 | 226.53 (202.66–253.20) | 210.02 (197.73–223.06) | 1.07 (0.95–1.22) | 0.272 |
| | Number of first-degree relatives in Sweden | | | | |
| | 0 | 653.31 (632.60–674.69) | 303.49 (291.31–316.17) | 2.00 (1.90–2.11) | <0.001 |
| | 1 | 316.71 (300.98–333.26) | 439.04 (416.33–462.99) | 0.67 (0.62–0.72) | <0.001 |
| | 2+ | 31.95 (27.35–37.32) | 63.67 (54.40–74.53) | 0.49 (0.39–0.61) | <0.001 |
| | **Multivariable analysis** | | | | |
| | Adjusted for age | N/A | N/A | 1.21 (1.16–1.26) | <0.001 |
| | Adjusted for age, sex, education, and calendar period of immigration | N/A | N/A | 1.17 (1.12–1.22) | <0.001 |
| | Adjusted for age, sex, education, calendar period of immigration, and smoking** | N/A | N/A | 1.20 (1.14–1.27) | <0.001 |
| Suicide | **Univariate analysis** | | | | |
| | Crude overall HR | 7.6 (6.3–9.1) | 11.0 (9.3–13.0) | 0.69 (0.54–0.89) | 0.004 |
| | Age at immigration | | | | |
| | 0–19 years | 2.52 (1.50–4.26) | 4.82 (2.90–7.99) | 0.39 (0.18–0.81) | 0.012 |
| | 20–39 years | 7.41 (5.61–9.81) | 9.27 (7.29–11.81) | 0.81 (0.56–1.18) | 0.274 |
| | 40–59 years | 12.26 (8.52–17.65) | 23.56 (17.95–30.91) | 0.54 (0.34–0.86) | 0.009 |
| | 60–79 years | 31.08 (20.83–46.38) | 14.59 (7.59–28.04) | 2.25 (1.04–4.84) | 0.039 |
| | 80 years or older | 0.00 incidence observed | 56.19 (14.05–224.66) | N/A | N/A |
| | Sex | | | | |
| | Male | 9.62 (7.66–12.09) | 13.50 (10.97–16.62) | 0.71 (0.52–0.98) | 0.037 |
| | Female | 5.50 (4.07–7.45) | 8.47 (6.50–11.03) | 0.65 (0.44–0.98) | 0.040 |
| | Education | | | | |
| | Primary | 7.25 (5.32–9.88) | 1.29 (9.46–15.97) | 0.65 (0.43–0.98) | 0.037 |
| | Secondary | 7.58 (5.81–9.90) | 14.91 (11.34–19.53) | 0.48 (0.33–0.70) | <0.001 |
| | University | 5.87 (3.47–9.91) | 7.09 (4.87–10.35) | 0.83 (0.43–1.59) | 0.572 |
| | Missing | 27.59 (13.80–55.16) | 7.65 (4.12–14.23) | 3.91 (1.54–9.91) | 0.004 |
| | Period of immigration | | | | |
| | 1991–1994 | 7.64 (6.19–9.43) | 12.15 (9.28–15.90) | 0.60 (0.43–0.85) | 0.004 |
| | 1995–1998 | 8.58 (5.65–13.03) | 10.30 (7.50–14.16) | 0.89 (0.52–1.50) | 0.656 |
| | 1999–2001 | 5.12 (2.44–10.73) | 10.52 (8.04–13.77) | 0.49 (0.22–1.08) | 0.076 |
| | Number of first-degree relatives in Sweden | | | | |
| | 0 | 11.82 (9.30–15.01) | 11.25 (9.10–13.92) | 1.09 (0.79–1.51) | 0.594 |
| | 1 | 7.28 (5.20–10.18) | 14.18 (10.55–19.06) | 0.50 (0.32–0.78) | 0.003 |
| | 2+ | 3.01 (1.82–5.00) | 6.16 (3.71–10.22) | 0.43 (0.21–0.90) | 0.025 |
| | **Multivariable analysis** | | | | |
| | Adjusted for age | N/A | N/A | 0.71 (0.56–0.92) | 0.008 |
| | Adjusted for age, sex, education, and calendar period of immigration | N/A | N/A | 0.63 (0.49–0.82) | <0.001 |
| | Adjusted for age, sex, education, calendar period of immigration, and smoking** | N/A | N/A | 0.68 (0.48–0.96) | 0.030 |

(*Continued*)

**Table 2.** (Continued)

| Cause of death | Analysis or characteristic | IR* (95% CI) for war migrants | IR (95% CI) for other European migrants | HR (95% CI) | *p*-Value for HR |
|---|---|---|---|---|---|
| Cardiovascular disease | **Univariate analysis** | | | | |
| | Crude overall HR | 120.2 (114.9–125.9) | 94.7 (89.6–100.2) | 1.20 (1.11–1.29) | <0.001 |
| | Age at immigration | | | | |
| | 0–19 years | 0.72 (0.27–1.92) | 2.89 (1.50–5.56) | 0.24 (0.07–0.81) | 0.021 |
| | 20–39 years | 13.76 (11.21–16.90) | 6.75 (5.08–8.95) | 1.69 (1.18–2.42) | 0.004 |
| | 40–59 years | 185.653 (169.07–203.86) | 121.86 (108.13–137.33) | 1.40 (1.20–1.54) | <0.001 |
| | 60–79 years | 1,511.44 (1,427.16–1,600.70) | 1,108.79 (1,028.73–1,195.08) | 1.26 (1.15–1.39) | <0.001 |
| | 80 years or older | 5,737.77 (4,864.72–6,767.51) | 6,461.30 (5,677.96–7,352.71) | 0.88 (0.71–1.08) | 0.223 |
| | Sex | | | | |
| | Male | 113.15 (105.87–120.92) | 100.87 (93.49–108.84) | 1.06 (0.96–1.18) | 0.260 |
| | Female | 127.39 (119.63–135.65) | 88.51 (81.57–96.05) | 1.35 (1.22–1.50) | <0.001 |
| | Education | | | | |
| | Primary | 270.69 (257.30–284.77) | 199.03 (186.49–212.42) | 1.23 (1.13–1.34) | <0.001 |
| | Secondary | 29.20 (25.49–33.45) | 47.38 (40.61–55.26) | 0.58 (0.47–0.72) | <0.001 |
| | University | 40.23 (32.93–49.14) | 21.54 (17.35–26.75) | 1.69 (1.25–2.28) | <0.001 |
| | Missing | 151.72 (112.91–203.88) | 68.12 (55.34–83.86) | 2.37 (1.65–3.40) | <0.001 |
| | Period of immigration | | | | |
| | 1991–1994 | 99.67 (94.03–105.64) | 119.89 (110.04–130.62) | 0.82 (0.74–0.91) | <0.001 |
| | 1995–1998 | 243.04 (224.69–262.90) | 110.07 (99.87–121.31) | 2.15 (1.89–2.43) | <0.001 |
| | 1999–2001 | 61.38 (49.56–76.02) | 61.73 (55.24–68.99) | 0.99 (0.78–1.26) | 0.920 |
| | Number of first-degree relatives in Sweden | | | | |
| | 0 | 251.69 (238.97–265.10) | 101.16 (94.24–108.60) | 2.33 (2.13–2.55) | <0.001 |
| | 1 | 86.67 (78.62–95.53) | 144.41 (131.64–158.42) | 0.56 (0.49–0.64) | <0.001 |
| | 2+ | 2.01 (1.081–3.73) | 11.50 (7.94–16.66) | 0.16 (0.08–0.34) | <0.001 |
| | **Multivariable analysis** | | | | |
| | Adjusted for age | N/A | N/A | 1.35(1.26–1.46) | <0.001 |
| | Adjusted for age, sex, education, and calendar period of immigration | N/A | N/A | 1.30 (1.20–1.40) | <0.001 |
| | Adjusted for age, sex, education, calendar period of immigration, and smoking** | N/A | N/A | 1.45 (1.29–1.62) [†] | <0.001 |
| Cancer | **Univariate analysis** | | | | |
| | Crude overall HR | 118.7 (113.3–124.3) | 81.7 (76.9–86.7) | 1.36 (1.26–1.47) | <0.001 |
| | Age at immigration | | | | |
| | 0–19 years | 5.04 (3.49–7.31) | 3.21 (1.73–5.97) | 1.73 (0.83–3.64) | 0.145 |
| | 20–39 years | 38.41 (33.97–43.44) | 15.46 (12.82–18.63) | 2.07 (1.65–2.60) | <0.001 |
| | 40–59 years | 306.60 (285.08–329.75) | 167.62 (151.38–185.60) | 1.60 (1.41–1.82) | <0.001 |
| | 60–79 years | 1,011.51 (943.00–1,085.00) | 841.32 (771.96–916.91) | 1.16 (1.03–1.30) | 0.009 |
| | 80 years or older | 1,220.80 (853.56–1,746.05) | 1,685.56 (1,308.73–2,170.87) | 0.74 (0.48–1.15) | 0.175 |
| | Sex | | | | |
| | Male | 126.02 (118.33–134.21) | 81.46 (74.85–88.65) | 1.42 (1.27–1.58) | <0.001 |
| | Female | 111.27 (104.03–119.01) | 81.89 (75.22–89.16) | 1.29 (1.16–1.44) | <0.001 |
| | Education | | | | |
| | Primary | 196.22 (184.87–208.26) | 130.79 (120.70–141.72) | 1.41 (1.27–1.56) | <0.001 |
| | Secondary | 65.56 (59.87–71.78) | 67.84 (59.66–77.16) | 0.89 (0.75–1.04) | 0.132 |
| | University | 82.12 (71.40–94.47) | 37.57 (31.89–44.27) | 1.90 (1.52–2.37) | <0.001 |
| | Missing | 248.28 (197.70–312.79) | 75.01 (61.54–91.44) | 3.53 (2.60–4.79) | <0.001 |
| | Period of immigration | | | | |
| | 1991–1994 | 106.70 (100.86–112.87) | 101.32 (92.30–111.22) | 1.03 (0.92–1.15) | 0.639 |
| | 1995–1998 | 194.67 (178.32–212.53) | 87.02 (78.01–97.08) | 2.21 (1.92–2.54) | <0.001 |
| | 1999–2001 | 76.00 (62.71–92.10) | 60.74 (54.30–67.94) | 1.25 (1.00–1.56) | 0.050 |
| | Number of first-degree relatives in Sweden | | | | |
| | 0 | 197.90 (186.65–209.82) | 83.68 (77.41–90.47) | 2.23 (2.02–2.46) | <0.001 |
| | 1 | 139.74 (129.42–150.88) | 129.91 (117.82–143.23) | 0.98 (0.86–1.11) | 0.717 |
| | 2+ | 8.64 (6.41–11.65) | 13.97 (9.98–19.55) | 0.60 (0.38–0.95) | 0.030 |
| | **Multivariable analysis** | | | | |
| | Adjusted for age | N/A | N/A | 1.45(1.34–1.56) | <0.001 |
| | Adjusted for age, sex, education, and calendar period of immigration | N/A | N/A | 1.43 (1.32–1.54) | <0.001 |
| | Adjusted for age, sex, education, calendar period of immigration, and smoking** | N/A | N/A | 1.27 (1.15–1.41) [††] | <0.001 |

*Incidence rates per 100,000 person-years.
**Country-specific smoking prevalence.
[†]Adjusted for baseline cardiovascular disease mortality.
[††]Adjusted for baseline cancer mortality.
IR, incidence rate; HR, hazard ratio; N/A, not applicable.

**Table 3. Mortality rates in migrants of the Balkan wars (exposed) compared to other European migrants (unexposed), by time since immigration.**

| Outcome measure | Time since immigration | | | |
|---|---|---|---|---|
| | 0–1 year | 2–4 years | 5–9 years | 10+ years |
| **Mortality (overall)** | | | | |
| IR* (95% CI) for exposed | 229.5 (209.8–251.1) | 270.4 (252.6–289.5) | 346.3 (330.2–363.2) | 443.1 (425.5–461.4) |
| IR (95% CI) for unexposed | 249.2 (230.8–269.0) | 244.2 (227.7–262.0) | 286.6 (271.4–302.6) | 393.0 (370.2–417.1) |
| HR** (95% CI) | 1.05 (0.89–1.24) | 1.05 (0.92–1.20) | 1.20 (1.09–1.32) | 1.35 (1.22–1.48) |
| p-Value for HR | 0.549 | 0.508 | <0.001 | <0.001 |
| **Suicide** | | | | |
| IR (95% CI) for exposed | 7.2 (4.3–11.9) | 7.5 (5.0–11.3) | 7.0 (5.0–9.8) | 8.3 (6.2–11.2) |
| IR (95% CI) for unexposed | 13.7 (9.9–18.9) | 10.9 (7.8–15.2) | 10.4 (7.8–13.9) | 9.5 (6.4–13.9) |
| HR** (95% CI) | 0.77 (0.33–1.79) | 0.56 (0.27–1.15) | 0.65 (0.35–1.20) | 0.82 (0.41–1.64) |
| p-Value for HR | 0.540 | 0.112 | 0.169 | 0.583 |
| **Death due to CVD†** | | | | |
| IR (95% CI) for exposed | 80.7 (69.4–93.9) | 99.4 (88.8–111.2) | 120.7 (111.3–130.8) | 147.4 (137.4–158.1) |
| IR (95% CI) for unexposed | 71.9 (62.3–82.9) | 82.8 (73.4–93.4) | 93.7 (85.2–103.1) | 132.2 (119.3–146.5) |
| HR** (95% CI) | 1.27 (0.91–1.77) | 1.38 (1.05–1.82) | 1.42 (1.16–1.73) | 1.55 (1.30–1.86) |
| p-Value for HR | 0.164 | 0.021 | <0.001 | <0.001 |
| **Death due to cancer††** | | | | |
| IR for exposed (95% CI) | 93.6 (81.4–107.7) | 93.9 (83.6–105.4) | 119.7 (110.4–129.8) | 141.9 (132.1–152.4) |
| IR for unexposed (95% CI) | 69.2 (59.8–80.1) | 66.8 (58.5–76.4) | 78.7 (70.9–87.3) | 115.8 (103.8–129.3) |
| HR** (95% CI) | 1.45 (1.09–1.93) | 1.10 (0.86–1.40) | 1.29 (1.08–1.55) | 1.29 (1.08–1.55) |
| p-Value for HR | 0.011 | 0.437 | 0.006 | 0.006 |

*Incidence rates per 100,000 person-years.

**Adjusting for sex, education, age at immigration, smoking (country-specific prevalence), and calendar period of immigration.

†Adjusted for baseline CVD mortality.

††Adjusted for baseline cancer mortality.

CVD, cardiovascular disease; HR, hazard ratio; IR, incidence rate.

## Morbidity

Table 4 shows morbidity rates among migrants of the Balkan wars and other European migrants stratified by background characteristics as well as crude, age-adjusted, and multivariable-adjusted hazard ratios for diagnosis of psychiatric disorders, PTSD, suicide attempt, CVD, and cancer.

Stratified analyses revealed that the elevated risk of CVD and cancer as well as the reduced risk of suicide attempt associated with being a migrant of the Balkan wars were fairly similar across strata of sex, age, and education. However, being male was associated with stronger risk elevations for PTSD (HR 13.28, 95% CI 10.46–16.84, $p < 0.001$, for males versus HR 5.65, 95% CI 4.81–6.64, $p < 0.001$, for females). Being a young or middle-aged migrant of the Balkan wars was also associated with a more pronounced risk increase for PTSD (HR 9.93, 95% CI 8.35–11.80, $p < 0.001$, for individuals aged 20–39 years; HR 15.83, 95% CI 11.45–21.87, $p < 0.001$, for individuals aged 40–59 years) (Table 4).

As compared to earlier immigration (in 1991–1994), later immigration (1999–2001) was associated with stronger risk elevations for both PTSD (HR 7.58, 95% CI 6.02–9.55, $p < 0.001$, for earlier migration versus HR 21.87, 95% CI 17.64–27.10, $p < 0.001$, for late immigration) and CVD morbidity (HR 1.22, 95% CI 1.17–1.26, $p < 0.001$, for earlier immigration versus HR 1.56, 95% CI 1.47–1.65, $p < 0.001$, for late immigration) (Table 4). With a trend test, overall

**Table 4. Rates of morbidities in migrants of the Balkan wars compared to other European migrants, across strata of sociodemographic and lifestyle-related characteristics.**

| Outcome measure | Analysis or characteristic | IR* (95% CI) for war migrants | IR (95% CI) for other European migrants | HR (95% CI) | p-Value for HR |
|---|---|---|---|---|---|
| **Overall psychiatric morbidity** | **Univariate analysis** | | | | |
| | Crude overall HR | 989.8 (973.7–1,006.3) | 753.2 (738.1–768.6) | 1.16 (1.13–1.19) | <0.001 |
| | Age at immigration | | | | |
| | 0–19 years | 726.24 (703.73–749.48) | 16.48 (882.68–951.57) | 0.67 (0.53–0.70) | <0.001 |
| | 20–39 years | 1,171.83 (1,145.15–1,199.14) | 648.69 (629.87–668.08) | 1.59 (1.53–1.65) | <0.001 |
| | 40–59 years | 1,227.00 (1,181.52–1,274.23) | 879.87 (840.27–921.33) | 1.30 (1.23–1.39) | <0.001 |
| | 60–79 years | 686.98 (630.08–749.01) | 670.58 (607.84–739.82) | 0.98 (0.86–1.12) | 0.811 |
| | 80 years or older | 454.63 (251.77–820.93) | 1,260.22 (934.63–1,699.25) | 0.39 (0.20–0.75) | 0.005 |
| | Sex | | | | |
| | Male | 907.45 (885.87–929.56) | 712.89 (692.32–734.07) | 1.11 (1.07–1.16) | <0.001 |
| | Female | 1,073.75 (1,050.03–1,098.01) | 794.13 (772.25–816.63) | 1.21 (1.17–1.25) | <0.001 |
| | Education | | | | |
| | Primary | 960.09 (933.89–987.01) | 807.85 (781.59–834.99) | 1.04 (1.00–1.09) | 0.061 |
| | Secondary | 1,016.42 (992.61–1,040.80) | 1,005.72 (971.42–1,041.24) | 0.94 (0.90–0.98) | 0.002 |
| | University | 885.36 (847.46–924.95) | 524.14 (501.28–548.04) | 1.51 (1.42–1.61) | <0.001 |
| | Missing | 1,813.60 (1,656.67–1,985.39) | 600.94 (559.85–645.03) | 3.02 (2.69–3.39) | <0.001 |
| | Period of immigration | | | | |
| | 1991–1994 | 895.41(877.76–913.42) | 775.16 (748.77–802.46) | 1.11 (1.07–1.16) | <0.001 |
| | 1995–1998 | 1,086.38 (1,045.60–1,128.75) | 752.74 (724.61–781.97) | 1.37 (1.30–1.45) | <0.001 |
| | 1999–2001 | 1,621.36 (1,552.21–1,693.59) | 734.80 (711.03–759.37) | 2.21 (2.09–2.33) | <0.001 |
| | Number of first-degree relatives in Sweden | | | | |
| | 0 | 1,010.76 (984.18–1,038.05) | 610.54 (592.86–628.76) | 1.50 (1.44–1.56) | <0.001 |
| | 1 | 1,282.23 (1,248.97–1,316.3) | 951.89 (917.11–987.99) | 1.21 (1.16–1.27) | <0.001 |
| | 2+ | 701.64 (678.31–725.76) | 954.17 (915.00–995.02) | 0.61 (0.58–0.65) | <0.001 |
| | **Multivariable analysis** | | | | |
| | Adjusted for age | N/A | N/A | 1.17 (1.14–1.20) | <0.001 |
| | Adjusted for age, sex, education, and calendar period of immigration | N/A | N/A | 1.28 (1.24–1.31) | <0.001 |
| | Adjusted for age, sex, education, calendar period of immigration, and smoking** | N/A | N/A | 1.19 (1.14–1.23) | <0.001 |
| **Post-traumatic stress disorder** | **Univariate analysis** | | | | |
| | Crude overall HR | 175.5 (168.9–182.2) | 18.6 (16.4–21.1) | 7.93 (6.95–9.06) | <0.001 |
| | Age at immigration | | | | |
| | 0–19 years | 56.95 (51.00–63.61) | 18.00 (13.85–23.39) | 2.57 (1.92–3.43) | <0.001 |
| | 20–39 years | 244.29(232.60–256.58) | 20.11 (17.07–23.70) | 9.93 (8.35–11.80) | <0.001 |
| | 40–59 years | 311.92(290.03–335.48) | 17.68 (12.92–24.20) | 15.83 (11.45–21.87) | <0.001 |
| | 60–79 years | 36.32(25.07–52.60) | 8.11 (3.38–19.49) | 4.03 (1.55–10.47) | 0.004 |
| | 80 years or older | N/A | N/A | N/A | N/A |
| | Sex | | | | |
| | Male | 180.94 (171.64–190.75) | 10.93 (8.67–13.77) | 13.28 (10.46–16.84) | <0.001 |
| | Female | 169.93 (160.89–179.48) | 26.35 (22.68–30.61) | 5.65 (4.81–6.64) | <0.001 |
| | Education | | | | |
| | Primary | 133.74 (124.40–143.78) | 16.25 (12.94–20.41) | 7.13 (5.60–9.08) | <0.001 |
| | Secondary | 190.78 (180.85–201.24) | 27.53 (22.49–33.69) | 6.07 (4.92–7.49) | <0.001 |
| | University | 169.45 (153.65–186.88) | 13.93 (10.65–18.24) | 10.10 (7.56–13.49) | <0.001 |
| | Missing | 664.25 (574.65–767.81) | 16.85 (11.10–25.60) | 40.01 (25.72–62.30) | <0.001 |
| | Period of immigration | | | | |
| | 1991–1994 | 143.21(136.41–150.36) | 17.67 (14.13–22.09) | 7.58 (6.02–9.55) | <0.001 |
| | 1995–1998 | 189.81(173.60–207.53) | 18.45 (14.54–23.40) | 9.50 (7.37–12.25) | <0.001 |
| | 1999–2001 | 421.90 (388.43–458.27) | 19.47 (15.97–23.734) | 21.87 (17.64–27.10) | <0.001 |
| | Number of first-degree relatives in Sweden | | | | |
| | 0 | 177.24 (166.57–188.60) | 14.44 (11.97–17.43) | 10.25 (8.40–12.51) | <0.001 |
| | 1 | 299.93 (284.52–316.18) | 29.37 (23.92–36.07) | 8.90 (7.19–11.02) | <0.001 |
| | 2+ | 58.03 (51.70–65.14) | 17.68 (13.11–23.84) | 2.52 (1.82–3.49) | <0.001 |
| | **Multivariable analysis** | | | | |
| | Adjusted for age | N/A | N/A | 8.15 (7.14–9.30) | <0.001 |
| | Adjusted for age, sex, education, and calendar period of immigration | N/A | N/A | 13.05 (11.36–14.98) | <0.001 |
| | Adjusted for age, sex, education, calendar period of immigration, and smoking** | N/A | N/A | 9.33 (7.96–10.94) | <0.001 |

*(Continued)*

**Table 4.** (Continued)

| Outcome measure | Analysis or characteristic | IR* (95% CI) for war migrants | IR (95% CI) for other European migrants | HR (95% CI) | p-Value for HR |
|---|---|---|---|---|---|
| **Suicide attempt** | **Univariate analysis** | | | | |
| | Crude overall HR | 54.8 (51.2–58.6) | 81.5 (76.7–86.5) | 0.62 (0.56–0.68) | <0.001 |
| | Age at immigration | | | | |
| | 0–19 years | 76.09 (69.14–83.73) | 107.20 (96.27–119.37) | 0.62 (0.54–0.72) | <0.001 |
| | 20–39 years | 50.08 (44.96–55.78) | 73.70 (67.64–80.30) | 0.65 (0.56–0.74) | <0.001 |
| | 40–59 years | 30.51 (24.21–38.43) | 81.08 (70.00–93.90) | 0.38 (0.29–0.50) | <0.001 |
| | 60–79 years | 16.84 (9.78–29.01) | 43.86 (30.08–63.96) | 0.35 (0.18–0.69) | 0.002 |
| | 80 years or older | 40.71 (5.73–289.00) | 56.21 (14.06–224.75) | 0.63 (0.06–7.14) | 0.711 |
| | Sex | | | | |
| | Male | 39.89 (35.66–44.62) | 66.84 (60.87–73.39) | 0.53 (0.46–0.62) | <0.001 |
| | Female | 69.78 (64.10–75.98) | 96.31 (89.03–104.19) | 0.68 (0.60–0.76) | <0.001 |
| | Education | | | | |
| | Primary | 60.53 (54.37–67.40) | 86.16 (78.03–95.14) | 0.58 (0.50–0.67) | <0.001 |
| | Secondary | 58.35 (52.99–64.25) | 130.71 (119.09–143.47) | 0.44 (0.38–0.50) | <0.001 |
| | University | 24.76 (19.19–31.96) | 38.72 (32.94–45.51) | 0.61 (0.45–0.83) | 0.002 |
| | Missing | 104.37 (72.97–149.27) | 61.42 (49.33–76.46) | 1.71 (1.12–2.60) | 0.012 |
| | Period of immigration | | | | |
| | 1991–1994 | 48.98 (45.07–53.22) | 81.99 (73.90–90.97) | 0.57 (0.48–0.65) | <0.001 |
| | 1995–1998 | 66.58 (57.29–77.38) | 86.66 (77.64–96.72) | 0.74 (0.61–0.89) | 0.001 |
| | 1999–2001 | 80.81 (67.04–97.41) | 77.17 (69.85–85.25) | 1.05 (0.85–1.30) | 0.649 |
| | Number of first-degree relatives in Sweden | | | | |
| | 0 | 39.08 (34.25–44.58) | 59.67 (54.39–65.45) | 0.63 (0.53–0.74) | <0.001 |
| | 1 | 59.08 (52.50–66.49) | 113.27 (101.98–125.79) | 0.48 (0.41–0.57) | <0.001 |
| | 2+ | 68.61 (61.69–76.31) | 108.68 (96.31–122.64) | 0.56 (0.47–0.66) | <0.001 |
| | **Multivariable analysis** | | | | |
| | Adjusted for age | N/A | N/A | 0.61 (0.55–0.66) | <0.001 |
| | Adjusted for age, sex, education, and calendar period of immigration | N/A | N/A | 0.58 (0.52–0.64) | <0.001 |
| | Adjusted for age, sex, education, calendar period of immigration, and smoking** | N/A | N/A | 0.57 (0.51–0.65) | <0.001 |
| **Cardiovascular disease** | **Univariate analysis** | | | | |
| | Crude overall HR | 1,276.3 (1,257.9–1,295.1) | 873.4 (857.1–890.0) | 1.37 (1.34–1.40) | <0.001 |
| | Age at immigration | | | | |
| | 0–19 years | 341.82 (326.58–357.79) | 257.99 (240.57–276.67) | 1.31 (1.20–1.42) | <0.001 |
| | 20–39 years | 1,013.24 (988.58–1,038.53) | 506.09 (489.59–523.15) | 1.72 (1.65–1.80) | <0.001 |
| | 40–59 years | 3,021.10 (2,945.59–3,098.55) | 1,906.94 (1,846.80–1,969.04) | 1.44 (1.38–1.50) | <0.001 |
| | 60–79 years | 6,776.02 (6,563.03–6,995.93) | 5,400.74 (5,193.73–5,615.99) | 1.23 (1.17–1.29) | <0.001 |
| | 80 years or older | 9,929.82 (8,616.87–11,442.83) | 12,258.48 (10,940.68–13,735.02) | 0.83 (0.70–1.00) | 0.051 |
| | Sex | | | | |
| | Male | 1,232.72 (1,207.23–1,258.76) | 864.93 (842.16–888.31) | 1.33 (1.28–1.38) | <0.001 |
| | Female | 1,320.38 (1,293.85–1,347.45) | 882.01 (858.87–905.77) | 1.41 (1.36–1.46) | <0.001 |
| | Education | | | | |
| | Primary | 1,650.26 (1,615.03–1,686.26) | 993.14 (963.76–1,023.42) | 1.65 (1.59–1.71) | <0.001 |
| | Secondary | 1,015.06 (991.23–1,039.47) | 992.47 (958.519–1,027.63) | 0.93 (0.92–1.01) | 0.082 |
| | University | 1,237.59 (1,192.15–1,284.76) | 702.53 (675.85–730.26) | 1.60 (1.52–1.69) | <0.001 |
| | Missing | 1,162.28 (1,042.19–1,296.21) | 661.16 (617.97–707.36) | 1.76 (1.55–2.01) | <0.001 |
| | Period of immigration | | | | |
| | 1991–1994 | 1,185.02 (1,164.45–1,205.96) | 980.60 (950.67–1,011.47) | 1.22 (1.17–1.26) | <0.001 |
| | 1995–1998 | 1,714.86 (1,662.60–1,768.76) | 868.24 (837.90–899.69) | 1.93 (1.85–2.03) | <0.001 |
| | 1999–2001 | 1,229.30 (1,170.26–1,291.31) | 786.70 (762.09–812.12) | 1.56 (1.47–1.65) | <0.001 |
| | Number of first-degree relatives in Sweden | | | | |
| | 0 | 1,816.64 (1,780.08–1,853.94) | 867.77 (846.53–889.55) | 1.98 (1.92–2.05) | <0.001 |
| | 1 | 1,643.75 (1,605.64–1,682.76) | 1,260.26 (1,219.73–1,302.14) | 1.19 (1.15–1.24) | <0.001 |
| | 2+ | 380.83 (363.82–398.63) | 420.32 (394.89–447.39) | 0.87 (0.80–0.94) | <0.001 |
| | **Multivariable analysis** | | | | |
| | Adjusted for age | N/A | N/A | 1.44 (1.41–1.47) | <0.001 |
| | Adjusted for age, sex, education, and calendar period of immigration | N/A | N/A | 1.45 (1.41–1.49) | <0.001 |
| | Adjusted for age, sex, education, calendar period of immigration, and smoking** | N/A | N/A | 1.39 (1.34–1.43) | <0.001 |

*(Continued)*

**Table 4.** (Continued)

| Outcome measure | Analysis or characteristic | IR* (95% CI) for war migrants | IR (95% CI) for other European migrants | HR (95% CI) | p-Value for HR |
|---|---|---|---|---|---|
| Cancer | Univariate analysis | | | | |
| | Crude overall HR | 233.5 (225.9–241.3) | 187.4 (180.1–195.0) | 1.21 (1.15–1.27) | <0.001 |
| | Age at immigration | | | | |
| | 0–19 years | 23.30 (19.60–27.68) | 16.40 (12.47–21.58) | 1.30 (0.94–1.81) | 0.117 |
| | 20–39 years | 133.02 (124.48–142.15) | 80.76 (74.41–87.66) | 1.48 (1.33–1.64) | <0.001 |
| | 40–59 years | 641.36 (609.43–674.96) | 449.84 (422.39–479.07) | 1.33 (1.22–1.44) | <0.001 |
| | 60–79 years | 1,399.62 (1,316.70–1,487.76) | 1,349.22 (1,257.90–1,447.18) | 1.04 (0.95–1.14) | 0.418 |
| | 80 years or older | 1,420.80 (1,015.20–1,988.45) | 1,623.98 (1,243.78–2,120.39) | 0.93 (0.60–1.43) | 0.733 |
| | Sex | | | | |
| | Male | 219.56 (209.29–230.34) | 174.13 (164.30–184.55) | 1.21 (1.12–1.31) | <0.001 |
| | Female | 247.56 (236.58–259.05) | 200.82 (190.16–212.09) | 1.21 (1.12–1.30) | <0.001 |
| | Education | | | | |
| | Primary | 310.46 (296.02–325.61) | 234.61 (220.88–249.19) | 1.35 (1.25–1.46) | <0.001 |
| | Secondary | 171.34 (161.95–181.27) | 185.05 (171.11–200.13) | 0.89 (0.80–0.98) | 0.014 |
| | University | 241.39 (222.34–262.06) | 153.85 (141.83–166.90) | 1.45 (1.29–1.63) | <0.001 |
| | Missing | 235.89 (185.99–299.18) | 126.84 (108.89–147.74) | 1.90 (1.43–2.52) | <0.001 |
| | Period of immigration | | | | |
| | 1991–1994 | 217.40 (208.96–226.18) | 214.48 (201.09–228.76) | 0.97 (0.90–1.05) | 0.449 |
| | 1995–1998 | 346.89 (324.67–370.62) | 198.96 (185.00–213.97) | 1.75 (1.59–1.94) | <0.001 |
| | 1999–2001 | 155.94 (136.30–178.41) | 155.54 (144.99–166.86) | 1.01 (0.87–1.18) | 0.904 |
| | Number of first-degree relatives in Sweden | | | | |
| | 0 | 356.36 (341.05–372.37) | 192.97 (183.27–203.18) | 1.81 (1.69–1.94) | <0.001 |
| | 1 | 301.98 (286.52–318.27) | 279.23 (261.10–298.63) | 1.02 (0.93–1.11) | 0.676 |
| | 2+ | 31.81 (27.22–37.18) | 54.44 (45.90–37.18) | 0.56 (0.44–0.71) | <0.001 |
| | Multivariable analysis | | | | |
| | Adjusted for age | N/A | N/A | 1.26 (1.19–1.32) | <0.001 |
| | Adjusted for age, sex, education, and calendar period of immigration | N/A | N/A | 1.20 (1.14–1.27) | <0.001 |
| | Adjusted for age, sex, education, calendar period of immigration, and smoking** | N/A | N/A | 1.16 (1.08–1.24) | <0.001 |

*Incidence rates per 100,000 person-years.

**Country specific prevalence.

HR, hazard ratio; IR, incidence rate.

psychiatric disorders ($p < 0.001$), PTSD ($p < 0.001$), and CVD ($p = 0.005$) increase significantly with the period of immigration.

Similarly, having a lower number of first-degree relatives in Sweden at immigration was associated with a more pronounced risk increase in overall psychiatric disorders (HR 1.50, 95% CI 1.44–1.56, $p < 0.001$, for no relatives; HR 1.21, 95% CI 1.16–1.27, $p < 0.001$, for 1 relative; HR 0.61, 95% CI 0.58–0.65, $p < 0.001$, for 2 or more relatives), PTSD (HR 10.25, 95% CI 8.40–12.51, $p < 0.001$, for no relatives; HR 8.90, 95% CI 7.19–11.02, $p < 0.001$, for 1 relative; HR 2.52, 95% CI 1.82–3.49, $p < 0.001$, for 2 or more relatives), and CVD (HR 1.98, 95% CI 1.92–2.05, $p < 0.001$, for no relatives; HR 1.19, 95% CI 1.15–1.24, $p < 0.001$, for 1 relative; HR 0.87, 95% CI 0.80–0.94, $p < 0.001$, for 2 or more relatives) (Table 4). With a trend test, overall psychiatric disorders ($p < 0.001$), PTSD ($p < 0.001$), and CVD ($p < 0.001$) decrease significantly with the number of first-degree relatives in Sweden.

Being a migrant of the Balkan wars was associated with a higher risk of psychiatric disorders compared to other European migrants during follow-up, with similar risk estimates in crude (HR 1.16, 95% CI 1.13–1.19, $p < 0.001$) and multivariable-adjusted models (HR 1.19, 95% CI 1.14–1.23, $p < 0.001$). Having been diagnosed with PTSD was also associated with being a migrant of the Balkan wars, with risk estimates increasing in multivariable-adjusted models (HR 9.33, 95% CI 7.96–10.94, $p < 0.001$) compared to the crude model (HR 7.93, 95% CI

6.95–9.06, $p < 0.001$). In contrast, being a migrant of the Balkan wars was associated with an overall reduced risk of suicide attempt, with similar risk estimates in crude (HR 0.62, 95% CI 0.56–0.68, $p < 0.001$) and multivariable-adjusted models (HR 0.57, 95% CI 0.51–0.65, $p < 0.001$) (Table 4).

Compared to other European migrants, being a migrant of the Balkan wars was further associated with an elevated risk of a clinical diagnosis of CVD, with similar risk estimates in crude (HR 1.37, 95% CI 1.34–1.40, $p < 0.001$) and multivariable-adjusted models (HR 1.39, 95% CI 1.34–1.43, $p < 0.001$). In addition, being a migrant of the Balkan wars was associated with an increased risk of cancer diagnosis, with relative risk estimates somewhat similar in the multivariable adjusted models (HR 1.16, 95% CI 1.08–1.24, $p < 0.001$) and the crude model (HR 1.21, 95% CI 1.15–1.27, $p < 0.001$) (Table 4).

Finally, both smoking-related cancer malignancies (overall HR 1.24, 95% CI 1.09–1.42, $p = 0.001$) and cancer malignancies not related to alcohol or smoking (overall HR 1.19, 95% CI 1.07–1.32, $p = 0.001$) were associated with being a migrant of the Balkan wars (S5 Table).

Sensitivity analysis excluding individuals with missing educational attainment had little influence on our main results (S6 Table).

When assessing time since migration, we found that the risk of overall psychiatric disorders was consistently increased among migrants of the Balkan wars throughout the follow-up period, with the risk of PTSD highly elevated during the first year (HR 30.98, 95% CI 17.51–50.83, $p < 0.001$) as well as 2–4 years after immigration (HR 18.93, 95% CI 12.16–29.46, $p < 0.001$). The risk of suicide attempt was decreased among migrants of the Balkan wars 5–9 years (HR 0.58, 95% CI 0.48–0.71, $p < 0.001$) and ≥10 years after immigration (HR 0.44, 95% CI 0.36–0.53, $p < 0.001$). Being a migrant of the Balkan wars was associated with a consistently increased risk of CVD throughout the follow-up period. Risk of cancer diagnosis was increased at all time points, although non-significant at 2–4 years after immigration (HR 1.13, 95% CI 0.96–1.33, $p < 0.001$) (Table 5).

## Discussion

This comprehensive follow-up study of more than 100,000 migrants of the Balkan wars to Sweden shows that compared to other European migrants to Sweden, being a war migrant is associated with considerable elevations in risks of cardiovascular- and cancer-related morbidities and mortality, as well as elevation in the risk of psychiatric disorders, specifically PTSD. Migration during the last years of the Balkan wars was associated with even greater risk elevations, as was having no first-degree relatives in Sweden at immigration.

Limitations of the study include lack of information about health status at the time of immigration. It is plausible that war migrants' health was worse at the time of immigration, as indicated by their increased risk of CVD during the first year of follow-up. It is also possible that war migrants diagnosed with late stage cancer opted to stay in Sweden for healthcare access rather than returning to the home country, causing bias in hazard rates. We also lack information about what populations immigrated from the countries of the Balkan wars at different time points. It is possible, for example, that soldiers immigrated later than civilians. Furthermore, we were not able to control for behavioral factors possibly contributing to increased risk of morbidity and mortality such as alcohol consumption, physical exercise, BMI, and diet. We also lack individual smoking history information, a known risk factor for many outcomes under study, and therefore derived prevalence estimates from national statistics from the country of emigration. In addition, the Swedish National Patient Register does not include psychiatric diagnoses made in primary care. Importantly, findings of this study apply to migrants of the Balkan wars, and caution must therefore be taken in generalizing our results to other migrant populations.

**Table 5. Rates of morbidities in migrants of the Balkan wars (exposed) compared to other European migrants (unexposed), by time since immigration.**

| Outcome measure | Time since immigration | | | |
|---|---|---|---|---|
| | 0–1 year | 2–4 years | 5–9 years | 10+ years |
| **Psychiatric disorders (overall)** | | | | |
| IR* (95% CI) for exposed | 554.2 (523.0–587.3) | 504.7 (479.9–530.8) | 946.9 (919.3–975.2) | 1,525.3 (1,490.6–1,560.7) |
| IR (95% CI) for unexposed | 556.7 (528.7–586.2) | 566.8 (541.0–593.8) | 747.4 (722.0–773.8) | 1,202.1 (1,159.7–1,245.9) |
| HR** (95% CI) | 1.23 (1.11–1.37) | 1.29 (1.18–1.42) | 1.34 (1.26–1.43) | 1.10 (1.04–1.16) |
| *p*-Value for HR | <0.001 | <0.001 | <0.001 | <0.001 |
| **Post-traumatic stress disorder** | | | | |
| IR (95% CI) for exposed | 90.9 (78.8–104.9) | 83.4 (73.7–94.3) | 150.4 (139.8–161.7) | 286.2 (272.0–301.1) |
| IR (95% CI) for unexposed | 6.1 (3.7–9.9) | 9.7 (6.8–13.8) | 20.4 (16.6–25.0) | 38.0 (31.3–46.0) |
| HR** (95% CI) | 30.98 (17.51–50.83) | 18.93 (12.16–29.46) | 9.73 (7.42–12.76) | 5.43 (4.31–6.83) |
| *p*-Value for HR | <0.001 | <0.001 | <0.001 | <0.001 |
| **Suicide attempt** | | | | |
| IR (95% CI) for exposed | 14.9 (10.5–21.2) | 51.7 (44.3–60.5) | 61.1 (54.5–68.4) | 66.5 (59.9–73.8) |
| IR (95% CI) for unexposed | 48.7 (41.0–57.9) | 69.2 (60.6–78.9) | 84.0 (75.9–92.9) | 123.3 (110.7–137.2) |
| HR** (95% CI) | 0.84 (0.52–1.35) | 0.89 (0.68–1.17) | 0.58 (0.48–0.71) | 0.44 (0.36–0.53) |
| *p*-Value for HR | 0.468 | 0.418 | <0.001 | <0.001 |
| **Cardiovascular diseases** | | | | |
| IR (95% CI) for exposed | 1,258.8 (1,211.2–1,308.5) | 848.2 (815.6–882.1) | 1,171.2 (1,140.2–1,203.0) | 1,659.6 (1,622.9–1,697.1) |
| IR (95% CI) for unexposed | 740.9 (708.5–774.8) | 719.2 (690.0–749.7) | 849.4 (822.2–877.6) | 1,249.2 (1,205.8–1,294.2) |
| HR** (95% CI) | 1.44 (1.32–1.56) | 1.27 (1.17–1.37) | 1.52 (1.43–1.61) | 1.37 (1.30–1.45) |
| *p*-Value for HR | <0.001 | <0.001 | <0.001 | <0.001 |
| **Cancer** | | | | |
| IR (95% CI) for exposed | 238.3 (218.2–260.2) | 188.4 (173.6–204.5) | 236.3 (223.0–250.4) | 255.3 (242.0–269.4) |
| IR (95% CI) for unexposed | 171.7 (156.5–188.3) | 165.0 (151.5–179.7) | 190.0 (177.6–203.2) | 224.7 (207.5–243.4) |
| HR** (95% CI) | 1.18 (1.00–1.40) | 1.13 (0.96–1.33) | 1.16 (1.03–1.31) | 1.26 (1.11–1.43) |
| *p*-Value for HR | 0.055 | 0.142 | 0.016 | <0.001 |

*Incidence rates per 100,000 person-years.

**Adjusting for sex, education, age at immigration, smoking (country-specific prevalence), and calendar period of immigration.

HR, hazard ratio; IR, incidence rate.

This large-scale epidemiological study with virtually complete register data on health outcomes of migrants of the Balkan wars immigrating to Sweden contributes to existing work by assessing mortality and major disease risk among both child and adult migrants from the same war-exposed geographic area. This study expands previous research with its extensive follow-up and by utilizing other migrants as a comparison group. Examining war migrants' health status compared to other migrants, rather than the native population, offers a unique opportunity to examine the health consequences of additional stressors brought on by war exposure.

Our results of greater risk of psychiatric morbidity among migrants of the Balkan wars extends previous research on the mental health of refugees compared to native populations in the host country [15,16], as our methodology included other European migrants as the comparison group, addressing the confounder of general migration-related stress. This permits firmer conclusions to be drawn about the effects of migration from a country affected by war. Our study also broadens earlier studies showing that war migrants have worse self-reported mental health [17] and more psychotropic drug prescriptions than other migrants [18]. The greater risk of psychiatric morbidity, specifically PTSD, found among war migrants in our study is likely explained by greater exposure to war-related traumatic events.

Interestingly, migrating during the last years of the Balkan wars, an indicator of longer war exposure duration and shorter time of follow-up, was associated with an increased risk of psychiatric morbidity, compared to migrating during the earlier years of the wars. Cumulative trauma exposure pre-migration [3] and longer duration of exposure to threat [19] have previously been associated with adverse mental health problems such as PTSD and depression among migrants.

This is, to our knowledge, the largest systematic investigation of major disease risk among a war-exposed cohort utilizing other migrants as the comparison group. Our findings gain support from studies showing similar results among samples of male refugees from various countries of origin in Sweden [6] and Finns who were forced to migrate from their home country after the Second World War [5]. We further found that later time period of migration was associated with greater risk increase of CVD compared to earlier migration, indicating that longer duration of exposure to war trauma may be associated with more elevated CVD risk. These results are in line with a recent study finding that exposure to traumatic events increases the risk of endothelial dysfunction, a hallmark of CVD [20]. The increased risk of CVD events and CVD mortality among migrants of the Balkan wars is possibly explained by the cumulative physiological strain caused by residing in war-stricken areas. Indeed, during the 1991 Balkan war in Croatia, the incidence of and mortality from acute myocardial infarction increased greatly among civilians who remained there [21].

We further found that being a migrant of the Balkan wars was associated with an overall increased risk of being diagnosed with or dying from cancer. Migrants have in general been found to have *lower* cancer mortality rates compared to native populations, leading to the healthy migrant hypothesis [22]. Few studies, however, have focused specifically on migrants of war, relative to other migrants. The increased cancer risk in our data is possibly explained by different lifestyles, greater stressor exposure among migrants of the Balkan wars compared to other European migrants, or both. Exposure to high levels of psychological stress as well as repeated exposure over a long period of time have both been linked to weakening of the immune system [23]. However, to date, no consensus has been reached on whether stress also affects the incidence of cancer [24]. Psychological stress and trauma can also impact cancer and CVD risk indirectly through behaviors such as smoking and alcohol consumption [25]. Sub-analysis by cancer type, however, revealed that cancer risk was evident for both smoking-related cancers and cancers not related to tobacco or alcohol consumption. As smoking is a dominant risk factor for CVD, migrants' country-level smoking prevalence was also controlled for. We further compared cancer- and CVD-related death rates among residents of the countries of the Balkan wars and the other European countries under study. We found that while cancer mortality rates were similar, CVD rates were on average higher in the countries of the Balkan wars in 1990 (S3 Table). It is therefore possible that the increased CVD risk among migrants of the Balkan wars is partly explained by risk factors attributable to the country of origin.

Not having any first-degree relatives in Sweden at immigration was associated with greater risk elevations for CVD and psychiatric disorders, particularly PTSD, among migrants of the Balkan wars. Previous research has found social support, family connectedness, and allegiances to one's original culture to be sources of resilience for war migrants [26]. Among refugees and asylum seekers, loneliness has also been found to be associated with risk of PTSD and other severe mental illnesses [27]. Previous research has also found unaccompanied refugee children to be at greater risk of psychiatric morbidity and inpatient care than children who settle with their families [16]. The increased risk of morbidity among the migrants of the Balkan wars could be because they lack social support but also because they have lost relatives in the war; bereavement is indeed an established risk factor for psychiatric morbidity [3]. This increased

risk might also reflect the additional stress migrants of war are faced with knowing their family members are residing in areas of conflict.

Despite war-related traumatic experiences being an established risk factor for suicidal behavior [28], with rates increasing in Serbia and Montenegro among males during the wars [29], we found that being a migrant of the Balkan wars was associated with a *decreased* risk of suicide compared to other migrants. Previous studies on risk of suicide among migrants compared to population controls have found conflicting results, with some studies finding war migrants to be at increased suicide risk [5] and others finding substantially lower risk [30,31]. As rates of suicide among migrants have been found to be strongly linked with country-of-origin suicide rates [30], we retrieved the risk patterns of suicide in the countries of the Balkan wars under study, and found them to be, on average, similar to those in the home countries of the other European migrants (S4 Table). A possible explanation for the decreased suicide risk among migrants of the Balkan wars is that those experiencing mental health problems, such as depression, at the time of the wars were less likely to migrate. Conversely, those who migrated from war could represent individuals who maintained a high drive for survival despite adversity. It is also possible that lower rates of suicide among war migrants are due to increased surveillance, such as more frequent health check-ups, among the war migrant group.

More people than ever are affected by forced displacement as a result of atrocities such as war and human rights violations. Our results underscore that war migrants are a particularly vulnerable group in need of long-term health surveillance and treatment for psychiatric and somatic disorders, particularly those with extended stay in war-stricken areas and those immigrating alone. Policy makers should assess possible barriers war migrants face in accessing treatment and enhance both the availability and accessibility of social services to war migrants, particularly those who are alone in the host country.

## Supporting information

**S1 Study Protocol.**
(DOC)

**S1 Table. Number of migrants with regard to whether they are migrants of the Balkan wars (exposed) or other European migrants (unexposed).**
(DOCX)

**S2 Table. Prevalence of smoking in the year 2000 in Balkan war countries (exposed) and other European countries (unexposed) among individuals that were 15 years or older.**
(DOCX)

**S3 Table. Cancer- and cardiovascular-disease-related death rates in the year 1990 (age-standardized rates per 100,000 inhabitants) in Balkan war countries (exposed) versus other European countries (unexposed).**
(DOCX)

**S4 Table. Mean suicide rate (age-adjusted rate of suicide per 100,000 inhabitants) in the year 1990 in the Balkan war countries (exposed) and other European countries (unexposed).**
(DOCX)

**S5 Table. Smoking- and alcohol-related cancer incidence among migrants of the Balkan wars (exposed) versus other European migrants (unexposed).**
(DOCX)

**S6 Table. Mortality and morbidity rates among migrants of the Balkan wars (exposed) compared to other European migrants (unexposed), with regard to whether individuals with missing education level are included.**
(DOCX)

**S1 Text STROBE checklist.**
(DOCX)

## Author Contributions

**Conceptualization:** Emily A. Holmes, Unnur Anna Valdimarsdottir.

**Data curation:** Edda Bjork Thordardottir, Li Yin, Emily A. Holmes, Unnur Anna Valdimarsdottir.

**Formal analysis:** Edda Bjork Thordardottir, Li Yin.

**Funding acquisition:** Edda Bjork Thordardottir, Emily A. Holmes, Unnur Anna Valdimarsdottir.

**Investigation:** Unnur Anna Valdimarsdottir.

**Methodology:** Edda Bjork Thordardottir, Emily A. Holmes, Unnur Anna Valdimarsdottir.

**Project administration:** Paul Lichtenstein.

**Supervision:** Emily A. Holmes, Unnur Anna Valdimarsdottir.

**Writing – original draft:** Edda Bjork Thordardottir, Emily A. Holmes, Unnur Anna Valdimarsdottir.

**Writing – review & editing:** Edda Bjork Thordardottir, Li Yin, Arna Hauksdottir, Ellenor Mittendorfer-Rutz, Anna-Clara Hollander, Christina M. Hultman, Paul Lichtenstein, Wei-min Ye, Filip K. Arnberg, Fang Fang, Emily A. Holmes, Unnur Anna Valdimarsdottir.

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
