## [Decision Letter · Decision Letter 0]

20 Apr 2020

Dear Dr. Thordardottir,

Thank you very much for submitting your manuscript "Mortality and major disease risk among migrants of the 1991-2001 Balkan wars to Sweden" (PMEDICINE-D-19-03629) for consideration at PLOS Medicine. 

[LINK]

In light of these reviews, I am afraid that we will not be able to accept the manuscript for publication in the journal in its current form, but we would like to consider a revised version that addresses the reviewers' and editors' comments. Obviously we cannot make any decision about publication until we have seen the revised manuscript and your response, and we plan to seek re-review by one or more of the reviewers. 

We expect to receive your revised manuscript by May 11 2020 11:59PM. Please email us (plosmedicine@plos.org) if you have any questions or concerns.

We look forward to receiving your revised manuscript. 

Sincerely,

Emma Veitch, PhD

PLOS Medicine

On behalf of Clare Stone, PhD, Acting Chief Editor,

PLOS Medicine

plosmedicine.org

*Please revise your title according to PLOS Medicine's style. Your title must be nondeclarative and not a question. It should begin with main concept if possible. Please place the study design ("A randomized controlled trial," "A retrospective study," "A modelling study," etc.) in the subtitle (ie, after a colon). eg, "Mortality and major disease risk among migrants of the 1991-2001 Balkan wars to Sweden: register-based cohort study".

*At this stage, we ask that you include a short, non-technical Author Summary of your research to make findings accessible to a wide audience that includes both scientists and non-scientists. The Author Summary should immediately follow the Abstract in your revised manuscript. This text is subject to editorial change and should be distinct from the scientific abstract. Please see our author guidelines for more information: https://journals.plos.org/plosmedicine/s/revising-your-manuscript#loc-author-summary

*Please ensure that the study is reported according to the STROBE guideline, and include the completed STROBE checklist as Supporting Information. When completing the checklist, please use section and paragraph numbers, rather than page numbers. Please add the following statement, or similar, to the Methods: "This study is reported as per the Strengthening the Reporting of Observational Studies in Epidemiology (STROBE) guideline (S1 Checklist)."

*As the study is observational, it isn't straight forward to infer causality; in most of the manuscript text, this is appropriately presented but there are a few places where the authors mention "elevated/increased risk" or "reduced risk" which does seem to imply the association is causal and the language could be fine-tuned in such instances (ie "associated with increased risk"). 

Comments from the reviewers:

Reviewer #1: I confine my remarks to statistical aspects of this paper. 

One big issue is the use of linear regression. Since there are a lot of people from each of a bunch of countries, the assumption of independent errors that linear regression makes will be violated. One method of dealing with this is a multilevel model. In addition, this allows clearer use of both individual and country-level variables. 

In addition, 

Line 106-7 Do not categorize independent variables. In *Regression Modeling Strategies* Frank Harrell lists 11 problems with doing this and sums up "Nothing could be more disastrous". I wrote a blog post illustrating some of the problems : https://medium.com/@peterflom/what-happens-when-we-categorize-an-independent-variable-in-regression-77d4c5862b6c

Peter Flom

Reviewer #2: The paper addresses a relevant health problem and the data used together with the analytic strategy adds to the literature. The topic as well as the data-set itself is very complex and this is a good attempt to attack the issues at hand. To this reviewer's mind, there are some points that could be sharpened in the paper. Comments:

The background and the conclusion in the abstract should - as the methods part does - make clear that this is a comparison between different groups of immigrants.

The authors present complex models first and then later return to stratified analyses (but also there we see only adjusted models). I would much have preferred to from the beginning see more presentation of the crude data, e.g. starting with something like table 4 displaying incidence rates, HR:s in unadjusted and then adjusted models in strata as in table 4. Then proceed to the multivariate models.

Likewise, it would be easier to understand the data and follow the line of interpretation if we in the modeling could see what happens in different steps of the adjustment. This need not be covariate by covariate, but could be in "chunks" of co-variates. The present results-section does not convince the reader that the models presented are the adequate ones and does not permit assessment of effect-modification and alike phenomena. 

Persons with missing data on education is an extreme group judging from table 4. It is not clear in the methods part how the missing category was handled and if the choice of method here influence the results in a major way. Have the authors considered an imputation here or some more advanced strategy to find out what is happening?

The tables are very unclear about what a "rate" is. E.g. table 4 shows hazard ratios, not rates, the other tables show both incidence rates and hazard ratios. It is very difficult to find out the enumeration of the denominators for the rates. 

It is very good that we for many tables see both incidence rates and the hazard ratio. This in turn could lead to a part in the discussion to show what the absolute health impact is likely to be in say 1000 individuals during five years of observation.

The discussion could be better structured. The study, say the authors, "offers a unique opportunity to examine the health consequences of additional stressors brought on by war exposure". The critical question is then if the design, data, analysis and results reflect a causal pathway for this, or if the findings are a result of other things such as data handling, selection of who migrates, or exposures in the home-country. Some of the hazard ratios are quite low and could be the result of bias rather than causal. The discussion enters this discussion but only partly and fragmented. Mostly, the authors use a rather strong "causal language" - which admittedly could be used around psychiatric disorders but is more questionable for e.g. cancer. A clearer structure could probably also lead to a shortening, which would put the message across clearer.

In the conclusion, line 315-321 is a repetition of the main findings and unnecessary. What comes after are consequences of the findings and the interpretation, which is a good final note. Could more of this be "transplanted" to the abstract?

Reviewer #3: Comments to author

This is an important and well-performed register-based investigation on the risk of mortality and major disease risk among migrants of the 1991-2001 Balkan wars to Sweden compare to migrants from other European counties to Sweden during the same period. The strengths of the study includes the use of comprehensive registers and careful statistical analyses. The manuscript is well written and easy to read.

I have only minor comments:

METHODS

1. They authors should describe any work/effort validating the information source. How accurate is the information in this database, how complete are the data; are they equally complete throughout the study period? 

2. Line 85, ICD-10 for CVD diagnosis is I00-I99, not 100-199

3. Line 94 and Line 97, ICD-7 code of 150 is "esophagus"

4. Line 95, ICD-7 code for lung cancer 162-163, code 161 is a diagnosis of larynx, the author has defined it as a separate subtype (Line 98).

5. Line 97, ICD-7 code for liver cancer are 155 and 156

6. Line 107-108, please indicate first degree relatives in the methods; as the authors have mentioned in Table 4, siblings, parents, and children.

7. Please clarify how the rate of morbidities and mortalities was calculated, it was for per 100 000 inhabitants or per 100 000 person years. 

RESULTS

8. Please add a footnote to indicate incidence rates (IR) in Table 1, 2 and Supplementary Table 5, for per 100 000 inhabitants, or per 100 000 person years. 

DISCUSSIONS AND CONCLUSION 

9. It should be noted that cancer survival in Sweden overall is internationally favorable. The findings that some groups with low cancer incidence may have even a more favorable survival. Even the study has controlled by the variable of residence time and country-level cancer and related rates, the possibility of immigrants diagnosed with a cancer in the earlier stage may move back to their home country, while severely ill patients stay in host country, causing bias in Hazard rates.

10. Suggest how your findings are useful for health care workers (or policy makers)? What might they do with this information?

[LINK]

---

## [Decision Letter · Decision Letter 1]

25 Aug 2020

Dear Dr. Thordardottir,

Thank you very much for re-submitting your manuscript "Mortality and major disease risk among migrants of the 1991-2001 Balkan wars to Sweden: A register-based cohort study" (PMEDICINE-D-19-03629R1) for consideration at PLOS Medicine.

I have discussed the paper with editorial colleagues and it was also seen again by two reviewers. I am pleased to tell you that, provided the remaining editorial and production issues are fully dealt with, we expect to be able to accept the paper for publication in the journal.

[LINK]

Please let me know if you have any questions. Otherwise, we look forward to receiving the revised manuscript shortly. 

Sincerely,

Richard Turner, PhD

rturner@plos.org

Requests from Editors:

Noting the comments from our academic editor (below), we ask you to give some thought to how study participants are described in the text (including the title and abstract). In particular, the term "war migrants" seems unusual, and "displaced persons" or "persons displaced by war" would be preferred. We suggest explaining the legal status of study participants in Sweden early in your article, and then using a standard term throughout the text - one possibility would be to use an abbreviation (e.g., "asylum seekers/refugees", "AS/R"). 

To your data statement, please non-author contact details to the two sources mentioned for readers interested in inquiring about access to study data.

Please add a sentence, say, to your abstract quoting aggregate demographic details for study participants. 

At line 26, please begin the sentence "Our findings indicate that ..." or similar. 

Early in the methods section of your main text, please state whether or not the study had a protocol or prespecified analysis plan, and if so attach the relevant document as a supplementary file, referred to in the text. Please highlight analyses that were not prespecified. 

Noting "p<0.0001" at line 203 and elsewhere, please quote exact p values or "p<0.001" throughout the article.

Please remove the information on funding and competing interests from the end of the main text. In the event of publication, this information will appear in the article metadata, via information provided in the submission form. 

Throughout the paper, please add p values alongside 95% CI, where available. 

Are you able to add a URL, and accessed date, to reference 9?

Please move the STROBE checklist to a separate attached file and refer to this in the methods section of your main text (i.e., "See S1_STROBE_Checklist" or similar). 

Comments from academic editor:

a. Definitions: I would suggest using the terms asylum seekers, refugees and (economic) migrants. 

i. I believe all displaced persons from the Balkan war would be considered asylum seekers while some may have received refugee status during their time in Sweden. However, the authors need to clarify whether Sweden gave the Balkan ‘refugees’ a special status until the war was done and then they had to return (I am not sure if this is the case, but Germany did). 

ii. Displaced persons from other countries (they may be asylum seekers, refugees or migrants). 

b. I also agree with the referee who said “I would much have preferred to from the beginning see more presentation of the crude data, e.g. starting with something like table 4 displaying incidence rates, HR:s in unadjusted and then adjusted models in strata as in table 4. Then proceed to the multivariate models.”

Comments from Reviewers:

***Reviewer #1: 

The authors have addressed my concerns and I now recommend publication.

*** Reviewer #3:

Comments:

The authors have satisfactorily responded to all my questions and made the necessary changes to the manuscript. I have no additional comments.

***

[LINK]

---

## [Editor Report · Decision Letter 2]

29 Oct 2020

Dear Dr. Thordardottir, 

On behalf of my colleagues and the academic editor, Dr. Paul Spiegel, I am delighted to inform you that your manuscript entitled "Mortality and major disease risk among migrants of the 1991-2001 Balkan wars to Sweden: A register-based cohort study" (PMEDICINE-D-19-03629R2) has been accepted for publication in PLOS Medicine. 

PRODUCTION PROCESS

Before publication you will see the copyedited word document (within 5 business days) and a PDF proof shortly after that. The copyeditor will be in touch shortly before sending you the copyedited Word document. We will make some revisions at copyediting stage to conform to our general style, and for clarification. When you receive this version you should check and revise it very carefully, including figures, tables, references, and supporting information, because corrections at the next stage (proofs) will be strictly limited to (1) errors in author names or affiliations, (2) errors of scientific fact that would cause misunderstandings to readers, and (3) printer's (introduced) errors. Please return the copyedited file within 2 business days in order to ensure timely delivery of the PDF proof. 

If you are likely to be away when either this document or the proof is sent, please ensure we have contact information of a second person, as we will need you to respond quickly at each point. Given the disruptions resulting from the ongoing COVID-19 pandemic, there may be delays in the production process. We apologise in advance for any inconvenience caused and will do our best to minimize impact as far as possible.

PRESS

PROFILE INFORMATION

Thank you again for submitting the manuscript to PLOS Medicine. We look forward to publishing it. 

Best wishes, 

Richard Turner, PhD

Senior Editor 

PLOS Medicine

plosmedicine.org